# Ultrastructure and Transcriptome Analysis Reveal Sexual Dimorphism in the Antennal Chemosensory System of *Blaptica dubia*

**DOI:** 10.3390/insects16101024

**Published:** 2025-10-03

**Authors:** Yu Zhang, Liming Liu, Haiqi Zhao, Jiabin Luo, Lina Guo

**Affiliations:** 1College of Animal Science, Shanxi Agricultural University, Taigu 030801, China; 15183537865@163.com (Y.Z.); sxaudkyllm@163.com (L.L.); zzhzqz13579@163.com (H.Z.); 18435051341@163.com (J.L.); 2Shanxi Key Laboratory of Animal Genetic Resource Discovery & Precision Breeding, Taigu 030801, China

**Keywords:** *Blaptica dubia*, sexual dimorphism, scanning electron microscope, transcriptome sequencing, chemosensory-related genes

## Abstract

**Simple Summary:**

*Blaptica dubia* relies on its chemosensory system to perform essential behaviors such as mate localization and environmental interactions. This study employed integrated morphological and transcriptomic analyses to investigate sexual dimorphism in this system. The results revealed significant differences between males and females in both sensilla morphology and gene expression profiles. These disparities suggest that the two sexes may have evolved distinct chemosensory strategies, providing new perspectives for understanding adaptive evolution in Blattodea insects.

**Abstract:**

This study distinguished male and female individuals by wing morphology (males with long wings, females with short wings) and investigated sexual dimorphism in the chemosensory system of *Blaptica dubia* through integrated ultrastructural and transcriptomic analyses. Scanning electron microscopy (SEM) was used to characterize the type, number, and distribution of antennal sensilla, while Illumina HiSeq sequencing, Gene Ontology/Kyoto Encyclopedia of Genes and Genomes (GO/KEGG) annotation, and Quantitative Real-time Reverse Transcription Polymerase Chain Reaction (qRT-PCR) validation were employed to analyze sex-specific gene expression profiles. Both sexes exhibited Böhm’s bristles, chaetic, trichoid, and basiconic sensilla. Males showed significantly more chaetic sensilla on the pedicel and longer type I/II chaetic sensilla on the flagellum, whereas females had longer ST2 sensilla. Basiconic sensilla were predominantly flagellar-distributed and more abundant/longer in males. No sexual differences were observed in Böhm’s bristles. Transcriptomics revealed 5664 differentially expressed genes (DEGs) (2541 upregulated; 3123 downregulated), enriched in oxidation-reduction, extracellular space, lysosome, and glutathione metabolism. KEGG analysis identified five key pathways: lysosome, glutathione metabolism, cytochrome P450-mediated xenobiotic/drug metabolism, and ascorbate/aldarate metabolism. Among 11 chemosensory-related DEGs, chemosensory proteins (CSPs) and odorant binding proteins (OBPs) were downregulated in males, while gustatory receptors (GRs), olfactory receptors (Ors), and ionotropic receptors (IRs) were upregulated. These results demonstrate profound sexual dimorphism in both antennal sensilla morphology and chemosensory gene expression, suggesting divergent sex-specific chemical communication strategies in *Blaptica dubia*, with implications for understanding adaptive evolution in Blattodea.

## 1. Introduction

Insects possess various types of cuticular sensory structures called sensilla, which are utilized for environmental perception due to their direct connection with the nervous system [1]. As essential sensory organs in insects, the antennae, densely covered with sensory sensilla on their surface, play a critical role in environmental perception, including olfactory signals, thermal variations, humidity gradients, and CO_2_ concentrations [2,3]. The insect chemosensory system, predominantly consisting of the olfactory and gustatory systems, serves as an essential physiological foundation that enables insects to detect environmental chemical cues and regulate behavioral outputs, critically mediating key biological processes, including foraging, oviposition site selection, and avoidance of noxious stimuli [4,5,6,7,8].

The sensory input from insect antennae represents a primary neural pathway for environmental signal processing in the central nervous system [9]. Chemoreception is mediated by a conserved suite of proteins that includes odorant-binding proteins (OBPs), chemosensory proteins (CSPs), odorant receptors (ORs), gustatory receptors (GRs), and ionotropic receptors (IRs) [6]. In *Periplaneta americana*, the chemosensory system has evolved remarkable adaptive sexual dimorphism. This is specifically manifested through morphological differentiation in sensilla structure: adult males possess approximately twice the number of olfactory sensilla compared to conspecific females, and their specialized antennal morphology (evidenced by differentiation in sensilla types and distribution patterns) shows significant correlation with mate-seeking behaviors [10,11].

The *Blaptica dubia*, also known as the South American Dubia roach or orange-spotted cockroach, is a large sexually dimorphic cockroach species [12]. In recent years, it has been widely utilized as live feed for experimental animals due to its high nutritional value, large body size, ease of captive breeding, and short growth cycle [13]. However, it remains unclear whether its chemosensory system exhibits sex-specific differences similar to those of the *P. americana*, or how the ultrastructure characteristics of its antennae and the expression profiles of chemosensory-related genes regulate behavioral differentiation between the sexes.

To systematically investigate the ultrastructural characteristics and sexual dimorphism of antennal sensilla in *Blaptica dubia*, we employed scanning electron microscopy (SEM) to obtain high-resolution morphological data on the major sensillar types of adult males and females. High-throughput transcriptome sequencing was subsequently used to quantify sex-biased expression of chemosensory genes in antennal tissues. By integrating morphological and molecular biological data, the research aims to reveal the sexual dimorphism of the *Blaptica dubia* chemosensory system and provides novel empirical insight into sex-specific adaptive strategies for chemical perception within the Blattodea.

## 2. Materials and Methods

### 2.1. Sample Collection

Adult males and females of *Blaptica dubia* were obtained from the experimental station of Shanxi Agricultural University.

### 2.2. Sample Preparation and Observation Methods

Ten sexually mature, apparently healthy *Blaptica dubia* cockroaches (10 males and 10 females) were randomly selected from rearing containers, with bilateral antennae from each individual included as experimental samples. Under a stereomicroscopic, antennae were carefully dissected at the scape with ultrafine forceps and a micro-scalpel to ensure structural integrity. The isolated antennae were sequentially washed five times with 70% ethanol to eliminate surface contaminants, with each washing step followed by air-drying a t room temperature (25 ± 1 °C). Samples were then fixed in 2.5% (*v*/*v*) glutaraldehyde in 0.1 M phosphate buffer (pH 7.2) for 24 h at 4 °C. After fixation, samples were rinsed three times (10 min each) with phosphate-buffered saline (PBS) to remove residual aldehyde. A graded ethanol series (30%, 50%, 70%, 80%, 90%, 95%, and 100%, *v*/*v*; 15 min per step) was employed for dehydration, with the 100% ethanol treatment being repeated twice. Dehydrated samples were critical-point dried using a JFD-320 freeze dryer (JEOL, Tokyo, Japan). Each sample was affixed to an aluminum stub with conductive carbon tape and sputter-coated with a 10 nm gold layer in an SBC-12 coater (KYKY Technology Co., Ltd., Beijing, China) to ensure optimal surface conductivity. The metallized samples were examined under high vacuum in a JSM-7800F field-emission scanning electron microscope (JEOL). Micrographs were acquired at 5–20 kV accelerating voltage [14,15,16].

Additionally, antennae were carefully excised through standard dissection protocols from three adult male and three adult female *Blaptica dubia* cockroaches, with bilateral antennae from each individual being utilized in the study. All excised antennal samples were rinsed three times in 75% (*v*/*v*) ethanol and subsequently air-dried at room temperature (25 ± 1 °C). The prepared antennae were then mounted onto white filter paper and imaged in their entirety using a KS-X1500S 3D digital super-depth microscope (Nanjing Kaishimai Technology Co., Ltd., Nanjing, China).

### 2.3. Image Processing

SEM micrographs were processed in Adobe Photoshop 2024 (Adobe Systems, San José, CA, USA) to optimize global contrast and uniform background without altering original morphological features. Antennal measurements were conducted using ImageJ software 1.54g (National Institutes of Health, USA).

### 2.4. Statistical Analysis

The total antennal length was measured using six antennae per sex, and the dimensions of each sensillum type were determined by measuring ten distinct structures from different antennal segments. Independent samples *t*-tests were employed for intergroup comparisons, and all statistical analyses were performed using SPSS 27 software. All datasets underwent normality assessment via the Shapiro–Wilk test, with significance levels exceeding 0.05, confirming compliance with the normality assumption. The measured data are expressed as mean ± standard error.

### 2.5. Total RNA Extraction and Quality Assessment

A total of 60 sexually mature *Blaptica dubia* adults (30 females and 30 males) were cold-anesthetized on an ice-covered aluminum foil stage. Antennae were rapidly dissected at the scape with sterile fine forceps, immediately transferred to 2 mL cryovials, flash-frozen in liquid nitrogen, and stored at −80 °C for subsequent RNA extraction. Total RNA was extracted from pooled antennal samples (10 antennae per biological replicate) using TRIzol reagent, yielding three independent replicates for each sex (A-1 to A-3 for females and B-1 to B-3 for males. RNA integrity was assessed using NanoDrop 2000 spectrophotometry (A260/280 ratio ≥1.8) (Thermo Fisher Scientific Inc., Waltham, MA, USA) and Agilent 2 100 Bioanalyzer (RIN >7.0) (Agilent Technologies, Santa Clara, CA, USA).

### 2.6. Transcriptomic Library Construction and Sequencing Protocol

Eukaryotic mRNA was enriched from quality-controlled RNA using oligo(dT) magnetic beads, fragmented, and reverse-transcribed into cDNA with random hexamers. Following second-strand synthesis, the double-stranded cDNA was purified, end-repaired, adenylated, and ligated to adapters. After size selection, the second strand was digested with USER enzyme, and the libraries were amplified by PCR. Final libraries were quality-checked and sequenced on the Illumina NovaSeq™ 6000 platform (PE150). High-quality RNA samples were subsequently submitted to LC-Bio Technology Co., Ltd. (Hangzhou, China) for library construction and Illumina paired-end sequencing. Raw data are available in the NCBI SRA under accession number PRJNA1288474 (see Appendix A).

### 2.7. qRT-PCR Validation Analysis

The RNA samples obtained from transcriptome sequencing were reverse-transcribed into cDNA using the PrimeScript RT Reagent Kit (Takara Bio Inc., Shiga, Japan). The resulting cDNA was then used as a template for quantitative PCR analysis with TB Green Premix Ex Taq (Tli RNaseH Plus) (Takara Bio Inc., Shiga, Japan) following the manufacturer’s protocol.

The 20 μL qPCR reaction mixture contained 10 μL TB Green^®^ Premix Ex Taq™ (Tli RNaseH Plus), 0.7 μL each of forward and reverse primers (10 µmol/L), 2 μL of cDNA, and 6.6 μL of nuclease-free water. The thermal cycling conditions consisted of initial denaturation at 95 °C for 3 min, 40 cycles of denaturation at 95 °C for 10 s, annealing at 58 °C for 30 s, and extension at 72 °C for 15 s. Melt curve analysis was performed under the following conditions: a cycle of decomposition steps (58 °C—1 min—followed by 0.5 °C up to 95 °C for 10 s). The Blaptica dubia arginine kinase (ArgK) gene was selected as the internal reference gene [17]. Three biological replicates, each with three technical replicates, were analyzed. Primer sequences are provided in Table 1.

## 3. Results

### 3.1. Morphological Characteristics of Blaptica dubia Antennae

The antennae of both male and female *Blaptica dubia* exhibit a slender, filiform structure comprising three morphologically distinct regions: the proximal scape, followed by the pedicel, and the multi-segmented flagellum [18] (Figure 1). Sexual dimorphism was observed in scape morphology: males exhibited significantly longer (males: 1.24 ± 0.05 mm; females: 1.16 ± 0.04 mm) yet more slender scapes. The pedicel displays a characteristic bipartite structure with bulbous termini and a subtle medial constriction, demonstrating significant size dimorphism (males: 0.49 ± 0.01 mm; females: 0.45 ± 0.01 mm). Total flagellar length differed markedly between sexes (males: 17.53 ± 0.32 mm; females: 12.97 ± 0.21 mm), though females possessed a significantly elongated first flagellomere (males: 0.41 ± 0.01 mm; females: 0.80 ± 0.02 mm) (Table 2). Notably, both sexes possess specialized cuticular scales on the 10th flagellomere (Figure 2), functioning to reinforce antennal structure.

### 3.2. Types of Antennal Sensilla in Blaptica dubia

Sexual dimorphism in *Blaptica dubia* antennae extends beyond gross morphology to include in the number and distribution of sensilla. SEM identified four main types of sensilla on their antennae: Böhm’s bristles, chaetic sensilla, trichoid sensilla, and basiconic sensilla, with distinct sex-specific patterns in their abundance and arrangement.

#### 3.2.1. Böhm’s Bristles

Böhm’s bristles were localized to the scape in both female and male antennae, predominantly clustered at the swollen basal region connecting to the head (Figure 3A,B). The pedicel showed minimal bristles distribution (Figure 3C,D). No significant quantitative difference in Böhm’s bristle numbers was observed between sexes.

#### 3.2.2. Chaetic Sensilla

Chaetic sensilla were distributed across all antennal segments (scape, pedicel, and flagellum) in both female and male *Blaptica dubia*. Males possessing significantly greater numbers on the pedicel compared to females. These sensilla were predominantly located on the flagellum and could be categorized into two distinct types: the shorter Type I (SCH_1_) and the longer Type II (SCH_2_). Both sensilla types were distributed circumferentially around the flagellum, with SCH_1_ and SCH_2_ localized to opposite poles of the flagellomeres (Figure 4A,B). Random measurements of 10 chaetic sensilla per type on the flagellum revealed significant sexual dimorphism in both sensillum types. Males exhibited greater lengths than females in Type I sensilla (SCH_1_) (males: 30–80 µm, mean 56.02 µm; females: 20–60 µm, mean 39.36 µm, *p* < 0.05) and type II sensilla (SCH_2_) (males: 120–150 µm, mean 133.97 µm; females: 90–130 µm, mean 106.65 µm, *p* < 0.05). Dimorphism was more pronounced in SCH_2_ (Table 3). Notably, the 10th flagellomere (F10) demonstrated a significant increase in sensilla density post-emergence (Figure 4A,B). Among all sensilla types, chaetic sensilla were the longest, exhibiting a rigid, spine-like morphology that contrasted sharply with the flexible structure of trichoid sensilla. Each sensillum was socketed within a cuticular depression containing a basal articular pit and displayed distinct longitudinal oriented striations extending toward the apex (Figure 4C,D).

#### 3.2.3. Trichoid Sensilla

In both female and male *Blaptica dubia*, trichoid sensilla were the most abundant antennal sensilla, showing sparse distribution versus dense flagellar clustering—sparse distribution on the scape and pedicel but dense clustering on the flagellum. Three morphologically distinct types of trichoid sensilla were identified in both male and female *Blaptica dubia.* Type I Trichoid Sensillum (ST_1_) emerges obliquely from the basal socket, extending distally before initiating curvature at 1/5 to 1/3 of the distance from the apex, with progressive tapering toward the terminal end. Type II Trichoid Sensillum (ST_2_) exhibits continuous curvature from the basal attachment, forming a smooth arc along their length with progressive distal tapering. In contrast, Type III sensilla (ST_3_) maintain straight alignment parallel to the cuticular surface, demonstrating uniform diameter reduction from base to apex (Figure 5). No significant sexual dimorphism was observed in trichoid sensilla abundance. While ST_1_ and ST_3_ exhibited no intersexual length differences, ST_2_ was significantly elongated in females (22.60 ± 2.70 µm) compared to males (15.21 ± 1.15 µm; *p* < 0.05) (Table 3).

#### 3.2.4. Basiconic Sensilla

In both female and male of *Blaptica dubia*, basiconic sensilla were shorter and stouter than the trichoid sensilla. These straight sensilla lacked distinct sockets, featuring broad base that tapered gradually to blunt tips containing a small terminal pore (Figure 6A,B). Basiconic sensilla were predominantly localized to the flagellum. Male individuals exhibited significantly longer basiconic sensilla (8.75 ± 0.52 µm) compared to females (6.65 ± 0.17 µm, *p* < 0.05), along with a significantly greater sensilla count (Table 3; Figure 6C,D).

### 3.3. Transcriptome Sequencing and Data Analysis

#### 3.3.1. Sequencing and Assembly of the Antennal Transcriptome in *Blaptica dubia*

High-throughput sequencing of *Blaptica dubia* antennal transcriptomes produced >5 GB high-quality clean data per sample after rigorous quality control. All samples showed excellent base quality and stable GC content (Table 4). De novo assembly with Trinity yielded 231,106 transcripts, with N50 of 1815 bp and GC 37.22%. We further obtained 116,290 non-redundant unigenes with strong continuity (N50 = 1047 bp) and consistent GC content (36.42%). These parameters confirm the high quality and reliability of our transcriptomic data for downstream analyses.

#### 3.3.2. Gene Functional Annotation

To obtain more comprehensive gene functional information, we annotated the 116,290 unigenes against six protein databases (NR, GO, KEGG, Pfam, SwissProt, and eggNOG) using BLASTx (E-value < 1 × 10^−5^). The NR database yielded the highest annotation rate (33,922 unigenes; 29.17%) (Table 5). BLASTx alignment against the NR database annotated 33,922 *Blaptica dubia* antennal unigenes. The highest sequence homology was observed with *Periplaneta americana* (22.65%) (Figure 7).

#### 3.3.3. Functional Classification of Differentially Expressed Genes

Gene Ontology (GO) annotation categorized 7948 unigenes into biological process (3729), cellular component (2059), and molecular function (2160). Oxidation-reduction process was the top term in biological process. For cellular component, ‘cytoplasm’ and ‘integral component of membrane’ were most abundant. In molecular function, ‘molecular function’ was the predominant term (Figure 8). Kyoto Encyclopedia of Genes and Genomes (KEGG) analysis assigned 1163 unigenes to 195 pathways. Only one pathway contained >40 unigenes, nine pathways had 20–30 unigenes each, and the majority (185 pathways, 94.9%) contained <10 unigenes (Table 6).

### 3.3.4. Enrichment Analysis of Differentially Expressed Genes

Among the top 25 most significantly enriched GO terms (Table 7), 10 mapped to biological process, 4 to cellular component, and 11 to molecular function. The term “oxidation-reduction process” (GO:0055114) showed the most significant enrichment in biological process, while “extracellular space” (GO:0005615) was most significantly enriched in cellular component. In molecular function, “oxidoreductase activity” (GO:0016491) demonstrated the highest enrichment significance. KEGG pathway enrichment assigned the DEGs to 195 pathways, with the top five significantly enriched pathways being Lysosome (map04142), Glutathione metabolism (map00480), Metabolism of xenobiotics by cytochrome P450 (map00980), Drug metabolism—cytochrome P450 (map00982), and Ascorbate and aldarate metabolism (map00053) (Figure 9).

#### 3.3.5. Screening Results of Chemosensory-related Differentially Expressed Genes

A total of 5664 DEGs were identified between male and female adult Dubia cockroaches, with 2541 up-regulated genes and 3123 down-regulated genes (Figure 10). Through manual keyword searches based on previous insect chemosensory research and DEG annotation information, we screened several chemosensory-related genes: 2 CSPs, 4 OBPs, 2 GRs, 1 OR, and 2 IRs. Notably, males displayed distinct expression profiles compared to females, characterized by complete downregulation of both CSPs, differential regulation of OBPs (with 1 upregulated, 3 downregulated), and consistent upregulation across all GRs, the single OR, and both IRs.

#### 3.3.6. Validation of Transcriptome Expression Levels

To validate the RNA-seq results, we performed comparative qRT-PCR analysis on 11 chemosensory-related DEGs identified in *Blaptica dubia* antennal transcriptomes. The results demonstrated concordant expression patterns for 9 out of these 11 genes between qRT-PCR and RNA-seq datasets, confirming the reliability of our transcriptome profiles (Table 8).

## 4. Discussion

Both male and female *Blaptica dubia* antennae possess the same types of sensilla, including Böhm’s bristles, chaetic sensilla, trichoid sensilla, and basiconic sensilla. Beyond the external sensilla structures described above, the antennal pedicel harbors Johnston’s organ—an internal mechanoreceptor playing a pivotal role in proprioception [19]. While the present study concentrates on the chemosensory functions of external sensilla, the potential synergistic mechanisms between Johnston’s organ as a crucial mechanosensory structure and the external chemosensory system merit further in-depth investigation in future research. Böhm’s bristles are primarily distributed on the scape, particularly in the antennal-head contact region, show significantly reduced occurrence on the pedicel, and are sparsely distributed in both male and female *Blaptica dubia*. Previous research has demonstrated that Böhm’s bristles function as proprioceptive mechanoreceptors, orchestrating precise antennal positioning and stabilization during insect flight [20]. Chaetic sensilla are morphologically segregated into two distinct types based on length differences: Type I and Type II. A pronounced sexual dimorphism was observed in *Blaptica dubia*, with males exhibiting significantly greater lengths in both Type I and Type II sensilla compared to females. These bifunctional sensory structures perform dual roles as both mechanoreceptors and chemoreceptors, capable of detecting contact pheromones and sex pheromones [21]. This sexual dimorphism in Type I and Type II sensilla length supports the hypothesis that the elongated Type I and Type II sensilla in males may represent an adaptive specialization, enhancing chemosensory capabilities to optimize mate-searching efficiency and thereby conferring competitive reproductive advantages. Based on morphological characteristics, the trichoid sensilla can be classified into three distinct types: Type I, Type II, and Type III. Previous studies on *Gromphadorhina brunneri* have shown that its sensilla trichodea possess exceptionally thin wall layers with distinctive dye permeability. This permeability is attributed to pore structures similar to those of known chemoreceptors, and the olfactory function of such sensilla has been experimentally confirmed in multiple Blattodea species and other insect taxa [22,23,24]. In *Blaptica dubia*, no significant intersexual differences occurred in ST_1_ and ST_3_ trichoid sensilla length. However, females exhibited significantly longer ST_2_ trichoid sensilla than males, suggesting that females may possess a more refined chemical detection system. Conversely, males possessed both a higher density and significantly greater length of basiconic sensilla compared to females. These sensilla are characterized by blunt-tipped shafts with small pores that function as conduits for odorant molecules. The presence of pore structures represents a well-documented ultrastructural characteristic of olfactory sensilla, serving as a diagnostic morphological feature for chemosensory function in insects [25].

High-throughput transcriptome sequencing has become an essential tool in insect molecular biology, enabling accurate de novo assembly of transcript libraries, particularly for non-model species lacking genomic references [26]. In this study, we performed transcriptome sequencing of male and female *Blaptica dubia* antennae to characterize sex-specific expression patterns, with particular emphasis on validating DEGs associated with chemosensory function. BLAST analysis against the NCBI non-redundant (NR) database identified significant homologous sequences for 33,922 of 116,290 unigenes (29.17%), indicating that the remaining ~71% represent putative novel transcripts that may underpin *Blaptica dubia*-specific biology and environmental adaptation [27]. The antennal transcriptome analysis of *Blaptica dubia* showed highest sequence similarity with *Periplaneta americana* (22.65%), indicating strong genetic homology between these two cockroach species. Although both cockroaches and termites belong to the order *Blattodea*, their sequence similarity was considerably lower (5.57–9.78%). This divergent may reflect distinct selective pressures imposed by their contrasting social (termites) and solitary (cockroaches) lifestyles [28].

GO enrichment analysis of DEGs between male and female *Blaptica dubia* antennae provided functional insights into their sexual dimorphism. Among the 3729 annotated DEGs, significant enrichment was observed across three major GO categories: biological process, cellular component, and molecular function. The most significantly enriched term in biological category was oxidation-reduction process (GO:0055114), whereas oxidoreductase activity (GO:0016491) was the predominant term in molecular function. This finding aligns with previous studies demonstrating that insect antennal aldehyde oxidases and dehydrogenases play a critical role in the degradation of various pheromones [29]. These findings may indicate significant intersexual divergence in pheromone degradation functionality, reflecting sex-specific adaptations in chemical communication. In the cellular component category, extracellular space (GO:0005615) was the most significantly enriched term. Given that insect antennae are the primary chemosensory organs, this enrichment likely reflects their critical role in processing chemical signals, including pheromone detection and clearance [30]. Given that OBPs and CSPs, small, soluble peptides abundant in sensillar lymph [31], are crucial for odorant recognition and transport [32], we hypothesize that *Blaptica dubia* exhibits sexually dimorphic expression patterns of these chemosensory proteins in its antennal tissues. KEGG pathway enrichment analysis of DEGs revealed cytochrome P450-associated metabolic pathways among the top five most significantly enriched pathways. Notably, two cytochrome P450-associated pathways demonstrated particularly strong enrichment: “Drug metabolism—cytochrome P450 (map00982)” and “Metabolism of xenobiotics by cytochrome P450 (map00980)”. These results corroborate the oxidation-reduction process identified in the GO analysis, consistent with the established role of cytochrome P450 enzymes in pheromone catabolism across diverse insect taxa [33]. The convergence of these findings provides compelling evidence for functional specialization in pheromone processing between male and female *Blaptica dubia* antennae, likely reflecting sex-specific adaptations in chemical communication.

OBPs and CSPs constitute the initial molecular components of the insect olfactory transduction cascade. These soluble proteins are predominantly localized in the sensillar lymph of olfactory sensilla. The detection process begins when volatile odor molecules permeate through cuticular pores and are captured by OBPs/CSPs. These carrier proteins then shuttle odorants to the dendritic membranes of olfactory receptor neurons (ORNs), where ligand–receptor interactions occur with either olfactory receptors (ORs) or ionotropic receptors (IRs), ultimately triggering action potential generation [34,35]. Transcriptomic profiling of *Blaptica dubia* revealed a consistent downregulation of OBP and CSP genes in male antennae compared to females. This sex-biased expression pattern correlates with the significant enrichment of extracellular space-associated genes (GO:0005615) identified in our GO analysis, suggesting reduced investment in soluble chemosensory proteins in males. Our findings suggest that male *Blaptica dubia* may have evolved a reduced dependence on conventional odorant detection systems, as evidenced by the downregulation of OBP/CSP genes and associated pathways. This sexual dimorphism in chemosensory gene expression likely reflects divergent ecological pressures and behavioral strategies between the sexes, with males potentially prioritizing mate localization, while females maintain broader olfactory sensitivity for host-finding and other ecological requirements. The observed molecular differences may underlie sex-specific adaptations in odorant processing and behavioral responses. Transcriptomic profiling of *Blaptica dubia* antennal tissues revealed a male-specific upregulation of three major chemosensory receptor gene families: GR, OR, and IR genes. This sexually dimorphic expression profile carries important functional implications for chemosensory processing in *Blaptica dubia*. Biochemical studies have established that ORs function as ligand-gated ion channels. Given their evolutionary homology with ORs, GRs are predicted to operate through similar ionotropic mechanisms. Furthermore, structural analyses have confirmed that IRs likewise mediate chemosensory signaling through ionic conductance [36]. Importantly, these ionotropic signaling pathways demonstrate superior temporal resolution, operating on millisecond to sub-millisecond timescales.

Male *Blaptica dubia* exhibit elongated Type I and Type II chaetic sensilla, increased density and length of basiconic sensilla, along with upregulation of IR, GR, and OR genes in their antennae. This morphological expansion may provide additional anchoring sites for membrane-bound receptor proteins (IRs, GRs, and ORs), thereby increasing receptor density, while the increased sensilla number could enhance signal capture efficiency by expanding the contact area with environmental chemical molecules [37]. These ion channel receptors operate on millisecond to sub-millisecond timescales. The integration of this ultrafast signal transduction mechanism with specialized sensilla structures significantly improves temporal resolution in signal processing. The three-dimensional integration of morphological, molecular, and temporal adaptations suggests that males may have evolved a unique “rapid-response” strategy, enabling millisecond-scale detection, transduction, and behavioral initiation in response to pheromonal signals. This provides critical advantages for time-sensitive behaviors such as mate localization and predator avoidance. This sensory specialization evolved for rapid and accurate recognition of conspecific signals is not unique to Blattodea. In moths, the “balanced olfactory antagonism” mechanism proposed by Baker (2008) similarly reveals how males achieve fast and precise mate localization in complex chemical environments through activation of specific receptors and avoidance of inhibitory signals [38]. This mechanism reflects convergent evolution under sexual selection pressure across species. In contrast, females appear to have evolved a distinct sensory adaptation pattern specialized for fine chemical discrimination: their significantly elongated ST_2_ trichoid sensilla may expand the distribution space for soluble proteins through increased surface area, while highly expressed OBP and CSP genes potentially enhance affinity and transport efficiency for diverse odor molecules, including host plant volatiles and oviposition site cues [39]. This morphology–molecular synergy may constitute a high-sensitivity chemical detection system that achieves precise resolution of subtle chemical gradients by lowering detection thresholds rather than increasing response speed. This refined perceptual strategy aligns with the female’s ecological role, enabling accurate identification of suitable oviposition sites, assessment of host plant quality, and detection of environmental threats through the coordinated action of elongated ST_2_ sensilla and highly expressed soluble proteins. As reviewed by Bruce and Pickett (2011), female insects exhibit fine discrimination capabilities towards blends of plant volatiles during oviposition host selection [40]. This highly sensitive chemosensory system is crucial for accurately assessing the suitability of oviposition sites, which aligns with the ‘high sensitivity’ strategy implied by the elongated ST_2_ sensilla and highly expressed OBP/CSP genes in female *Blaptica dubia* from our study. Insects are widely distributed and have adapted to diverse ecological and behavioral niches, which are partially defined by their sensory capabilities [41]. This adaptive divergence between male and female cockroaches may reflect niche partitioning—males specialize in “generalized detection” for rapid mate localization, while females excel in “specialized recognition” for fine chemical discrimination, collectively demonstrating the precise evolution of sensory systems under sexual selection pressure.

## 5. Conclusions

Through integrated morphological and transcriptomic analyses, this study reveals significant sexual dimorphism in *Blaptica dubia*, demonstrating distinct antennal sensilla structures and divergent chemosensory gene expression profiles between sexes. Morphological characterization reveals specialized sensilla in males that enhance chemosensory acuity, specifically elongated Type I and Type II chaetic sensilla, increased numbers of basiconic sensilla, and greater sensilla length, corresponding to their requirements for rapid chemical signal response. In contrast, females exhibit longer ST_2_ trichoid sensilla, indicative of enhanced chemical perception capabilities. These sexually dimorphic traits are interpreted as adaptations enabling rapid and accurate detection of chemical cues critical to male reproductive behavior. Transcriptomic profiling revealed sexually divergent gene expression underpinning distinct chemosensory strategies: males up-regulate IR, GR, and OR genes for rapid signal detection, whereas females preferentially express OBP and CSP genes for fine-tuned chemical discrimination. These findings elucidate sex-specific adaptations in insect chemosensation and advance our understanding of the evolutionary drivers of chemical perception. The multidisciplinary pipeline described here offers a transferable model for dissecting sensory adaptations across taxa.

## Figures and Tables

**Figure 1 insects-16-01024-f001:**
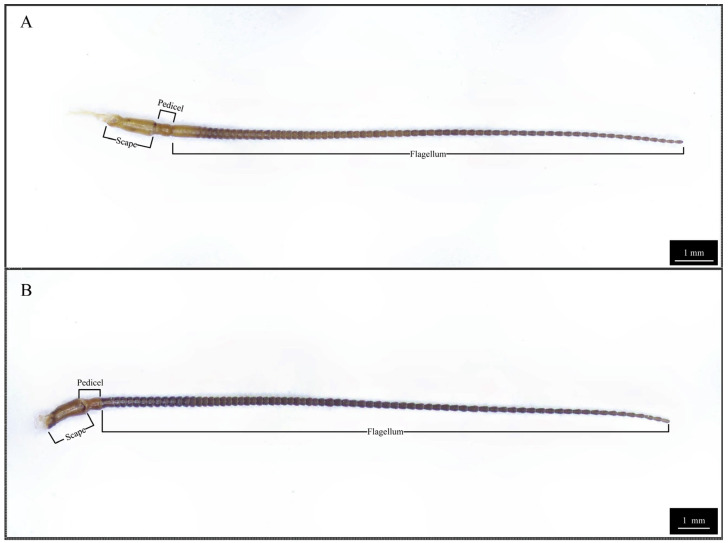
Three-dimensional optical microscopy of *Blaptica dubia* antennae. Note: (**A**) Female antenna; (**B**) Male antenna.

**Figure 2 insects-16-01024-f002:**
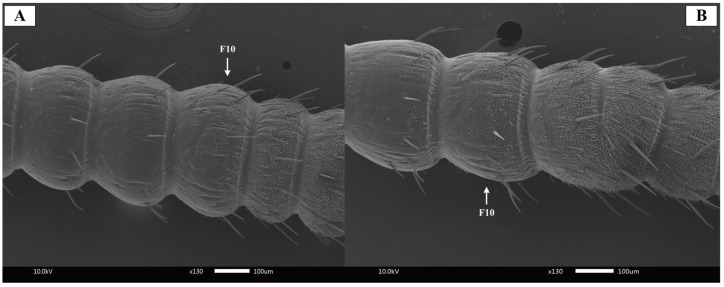
Scanning electron micrographs of *Blaptica dubia* antennal flagellum. Note: (**A**) Female antenna; (**B**) Male antenna; F10 = 10th flagellomere.

**Figure 3 insects-16-01024-f003:**
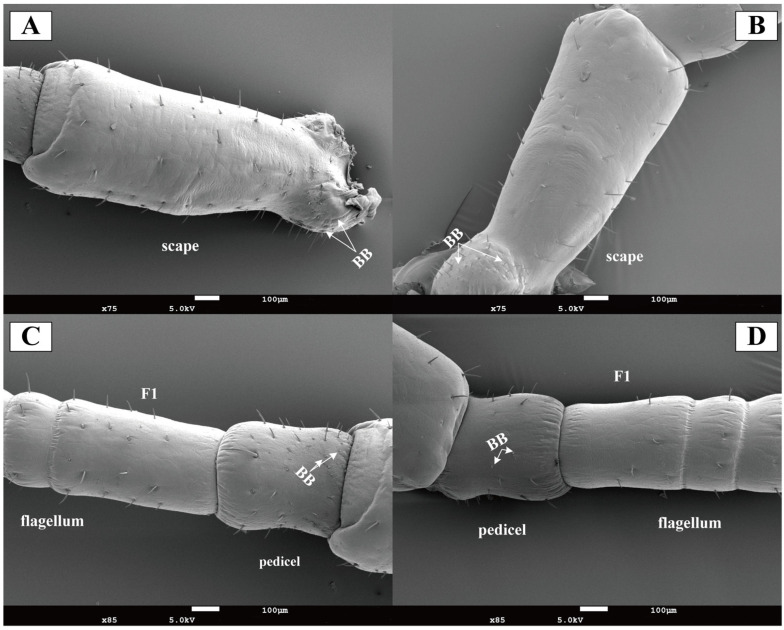
Scanning electron micrographs of *Blaptica dubia* antennal scape, pedicel, and first flagellomere. Note: (**A**) Female scape; (**B**) Male scape; (**C**) Female pedicel and flagellum; (**D**) Male pedicel and flagellum; F1 represents sensilla on the first flagellomere; BB indicates Böhm’s bristle.

**Figure 4 insects-16-01024-f004:**
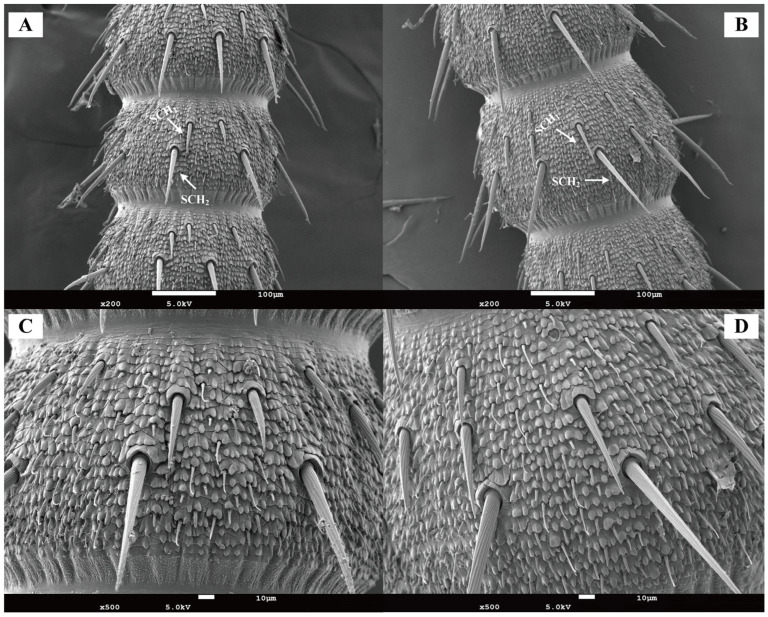
Scanning electron micrographs of chaetic sensilla in *Blaptica dubia.* Note: (**A**,**C**) show female chaetic sensilla; (**B**,**D**) display male chaetic sensilla. SCH1 denotes Type I chaetic sensilla, while SCH2 indicates Type II chaetic sensilla.

**Figure 5 insects-16-01024-f005:**
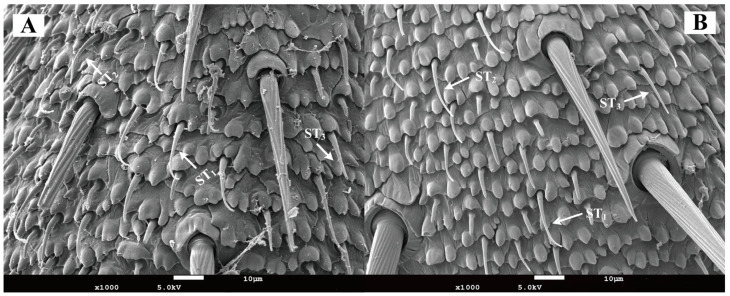
Scanning electron micrographs of trichoid sensilla on the antenna of *Blaptica dubia.* Note: (**A**) Female trichoid sensilla; (**B**) Male trichoid sensilla. ST_1_ represents Type I trichoid sensilla; ST_2_ represents Type II trichoid sensilla; ST_3_ represents Type III trichoid sensilla.

**Figure 6 insects-16-01024-f006:**
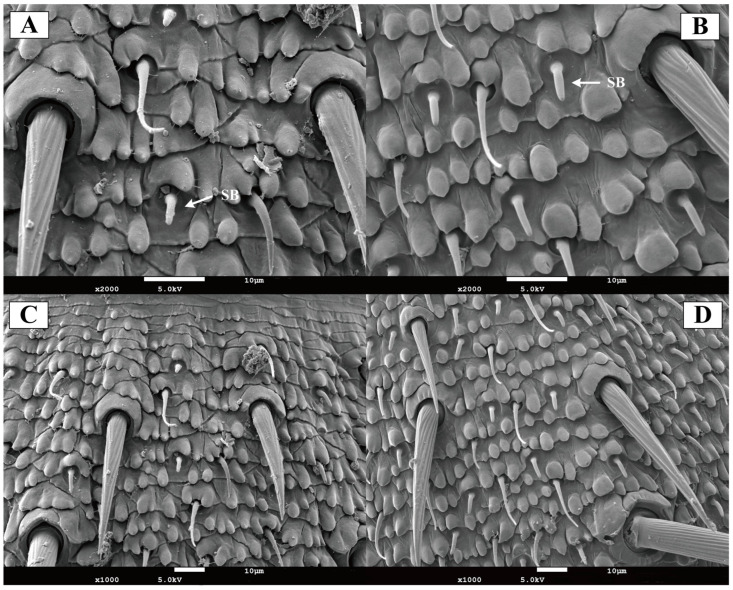
Scanning electron micrographs of basiconic sensilla on the antenna of *Blaptica dubia.* Note: (**A**,**C**) show female basiconic sensilla (SB); (**B**,**D**) show male basiconic sensilla (SB).

**Figure 7 insects-16-01024-f007:**
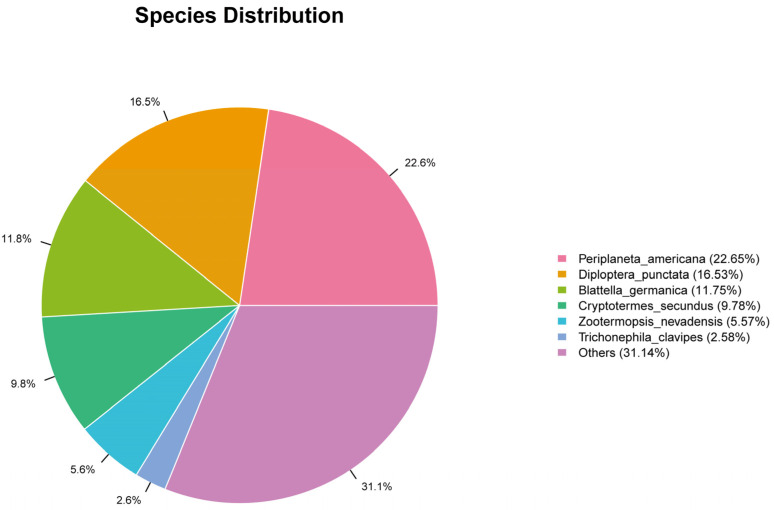
Distribution map of unigene in NR database.

**Figure 8 insects-16-01024-f008:**
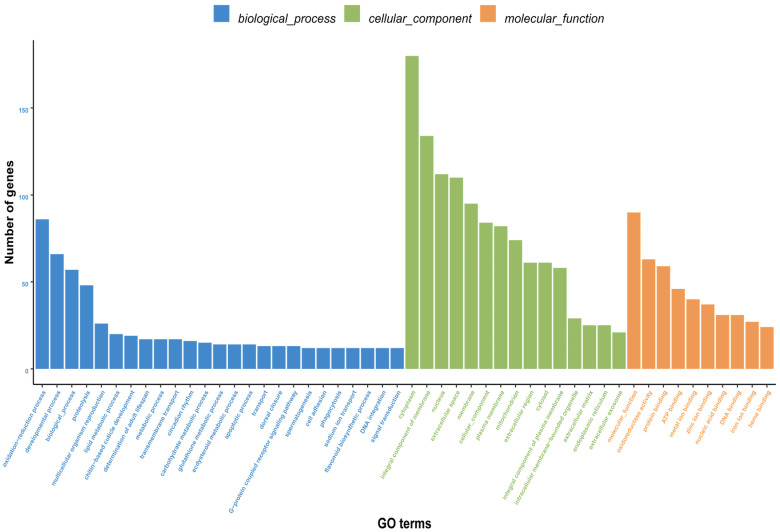
GO classification of differentially expressed genes (DEGs) between male and female *Blaptica dubia* antennae. Bar plot shows gene counts (vertical axis) across secondary GO terms (horizontal axis), color-coded by ontology: biological process (BP, blue), cellular component (CC, green), and molecular function (MF, red).

**Figure 9 insects-16-01024-f009:**
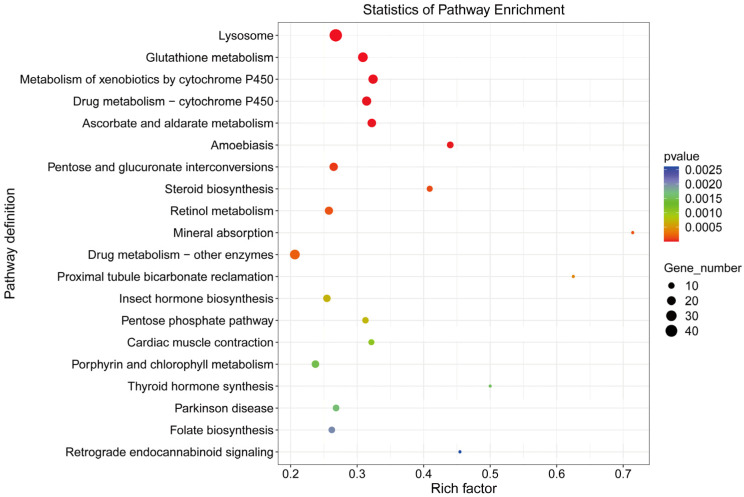
KEGG pathways of differentially expressed genes between male and female antennae in *Blaptica dubia*.

**Figure 10 insects-16-01024-f010:**
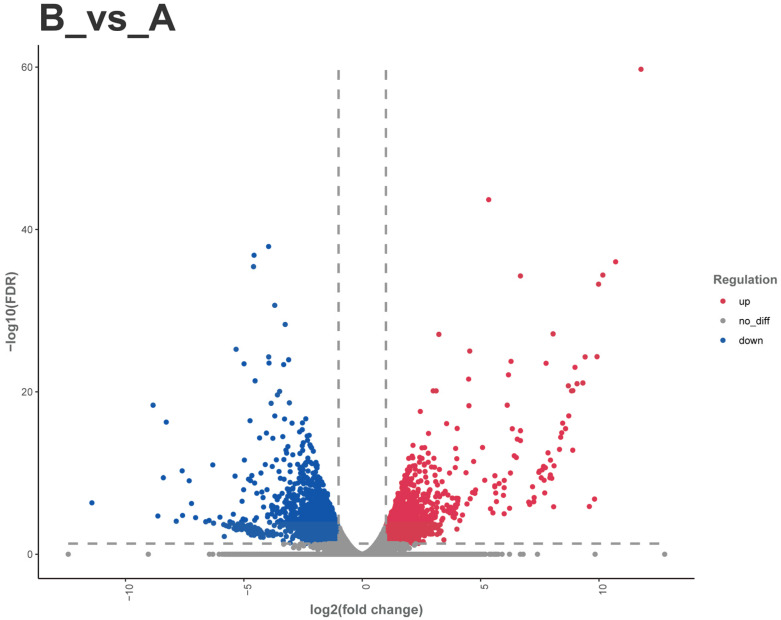
Volcano plot of differentially expressed genes between female and male *Blaptica dubias*. Note: (A) Female; (B) Male. Red dots: upregulated genes; blue dots: downregulated genes; gray dots: non-significantly differentially expressed genes.

**Table 1 insects-16-01024-t001:** Primer information.

Primer Name	Forward Primer Sequence (5′-3′)	Reverse Primer Sequence (5′-3′)
*ArgK*	CTCGTGTGGTGCAACGAAGA	GGTGGCTGAACGGGACTCT
*TRINITY_DN5543_c1_g1*	TCAGGGACGGACGCAACTA	CCGCCAGAAAGAAATGAGGTAG
*TRINITY_DN23994_c0_g1*	GCAGTAGACCGCAGACCTCA	ACGATCTTGGACCATTCCTTAG
*TRINITY_DN10322_c0_g1*	ATTCACGGAGTGCTTCTGGTC	TGGCGGTATCAATGCTTCA
*TRINITY_DN6009_c0_g1*	GTCAAGCAGGCGGCAAAGA	AGAGCAAGGTCGCAGGCATC
*TRINITY_DN8952_c1_g1*	AGTTGGATCACCTGGACGAA	AATGCACGCCATGTAGCC
*TRINITY_DN4687_c0_g1*	GTCAGCCCGAAACAGGAGT	TGCGATGTCACAGGAATCAG
*TRINITY_DN10421_c0_g1*	AGGGGAACGAGGAGGCTAT	CGACCCACGATGTGCTTTA
*TRINITY_DN8525_c2_g1*	CTCTTCGTTCCAAATAAACCG	ACCACAACAAAGTCTCCCAAA
*TRINITY_DN2213_c0_g3*	ACAAACTACATGAACGAAACGC	CAATCGCTGAAATTCCCAGT
*TRINITY_DN2362_c0_g1*	ATCGTTGTGGTTCGTCGTAGT	AGGAAGGAGATGAGGGTTGC
*TRINITY_DN54802_c0_g1*	ATGCGGCAATGACGAACG	ACAACACGACGGACACGGA

Note: The primer names are consistent with the gene names for which they were synthesized, and the italics follow the convention for gene nomenclature.

**Table 2 insects-16-01024-t002:** Lengths of antennal segments of *Blaptica dubia* adult in both sexes.

Antennal Segment		Length (mm)	
		Male	Famale
Scape		1.24±0.05	1.16 ± 0.04
Pedicel		0.49±0.01	0.45 ± 0.01 *
Flagellum	F1	0.41±0.01	0.80 ± 0.02 *
Flagellum	TFL	17.53±0.32	12.46 ± 0.31 *
Total		19.86±0.47	15.33 ± 0.62 *

Notes: F1 represents the first flagellomere; TFL represents the total flagellum length; Total represents the total length of the antennae. The data in the table represents the mean ± SE. Significant differences between males and females were determined by independent samples *t*-test. The asterisk (*) indicate significant difference between male and female (*p* < 0.05).

**Table 3 insects-16-01024-t003:** The type and length of antennal sensilla of *Blaptica dubia* adult in both sexes.

Type of Sensilla	Length (µm)	
	Male Adult	Famale Adult
SCH_1_	56.02 ± 3.28	39.36 ± 3.23 *
SCH_2_	133.97 ± 2.48	106.65 ± 3.78 *
ST_1_	16.03 ± 0.98	15.92 ± 0.85
ST_2_	15.21 ± 1.15	24.09 ± 2.25 *
ST_3_	18.46 ± 1.24	17.82 ± 1.11
SB	8.75 ± 0.52	6.65 ± 0.17 *

Notes: SCH_1_ represents Type I chaetic sensilla; SCH_2_ represents Type II chaetic sensilla; ST_1_ represents Type I trichoid sensilla; ST_2_ represents Type II trichoid sensilla; ST_3_ represents Type III trichoid sensilla; SB represents basiconic sensilla. The data in the table represent the mean ± SE. Significant differences between males and females were determined by independent samples *t*-test. The asterisk (*) indicate significant difference between male and female (*p* < 0.05).

**Table 4 insects-16-01024-t004:** Transcriptome sequencing results of antennal samples from male and female adult *Blaptica dubia*.

Sample	Raw Bases (G)	Clean Bases (G)	Q20 (%)	Q30 (%)	GC Content (%)
A-1	6.11G	5.91G	99.05	97.14	35.81
A-2	6.54G	6.32G	99.09	97.20	36.15
A-3	5.32G	5.17G	99.07	97.21	37.57
B-1	6.08G	5.86G	98.97	97.02	35.98
B-2	6.05G	5.82G	98.98	97.06	35.60
B-3	6.07G	5.82G	99.03	97.15	35.56

Notes: A represents the antennal sample of female *Blaptica dubia*, B represents the antennal sample of male *Blaptica dubia*.

**Table 5 insects-16-01024-t005:** Functional annotation results of the *Blaptica dubia* transcriptome.

Annotation Database	Number of Unigene	Percentage
All	116,290	100.00
NR	33,922	29.17
eggNOG	17,874	15.37
Pfam	16,775	14.43
GO	11,813	10.16
SwissProt	9643	8.29
KEGG	5874	5.05
TF	508	0.44

**Table 6 insects-16-01024-t006:** KEGG pathway analysis of differentially expressed genes in the antennae of female and male *Blaptica dubia*.

Pathway ID	Pathway Definition	Count
map04142	Lysosome	45
map00190	Oxidative phosphorylation	28
map00310	Lysine degradation	28
map00983	Drug metabolism—other enzymes	26
map00480	Glutathione metabolism	25
map04080	Neuroactive ligand-receptor interaction	24
map00980	Metabolism of xenobiotics by cytochrome P450	23
map04145	Phagosome	23
map00982	Drug metabolism—cytochrome P450	22
map03040	Spliceosome	20

**Table 7 insects-16-01024-t007:** GO enrichment analysis of DEGs in the antennae of female and male *Blaptica dubia*.

GO Category	Term Type	GO Code	Number	*p*-Value
extracellular space	CC	GO:0005615	110	5.55 × 10^−16^
extracellular region	CC	GO:0005576	61	8.56 × 10^−14^
oxidation-reduction process	BP	GO:0055114	86	3.06 × 10^−12^
oxidoreductase activity	MF	GO:0016491	63	1.79 × 10^−10^
multicellular organism reproduction	BP	GO:0032504	26	7.38 × 10^−10^
structural constituent of cuticle	MF	GO:0042302	20	4.88 × 10^−9^
oxidoreductase activity, acting on CH-OH group of donors	MF	GO:0016614	18	9.00 × 10^−9^
serine-type endopeptidase inhibitor activity	MF	GO:0004867	21	1.51 × 10^−8^
chitin binding	MF	GO:0008061	16	1.87 × 10^−7^
ecdysteroid metabolic process	BP	GO:0045455	14	2.05 × 10^−7^
extracellular matrix	CC	GO:0031012	25	2.20 × 10^−7^
flavin adenine dinucleotide binding	MF	GO:0050660	22	2.35 × 10^−6^
proteolysis	BP	GO:0006508	48	2.38 × 10^−6^
integral component of membrane	CC	GO:0016021	134	8.10 × 10^−6^
peptidoglycan binding	MF	GO:0042834	8	8.88 × 10^−6^
peptidoglycan catabolic process	BP	GO:0009253	7	9.94 × 10^−6^
N-acetylmuramoyl-L-alanine amidase activity	MF	GO:0008745	7	9.94 × 10^−6^
lipid metabolic process	BP	GO:0006629	20	0.00
glutathione metabolic process	BP	GO:0006749	14	0.00
circadian rhythm	BP	GO:0007623	16	0.00
UDP-glycosyltransferase activity	MF	GO:0008194	12	0.00
chitin-based cuticle development	BP	GO:0040003	19	0.00
structural constituent of chitin-based larval cuticle	MF	GO:0008010	12	0.00
oxidoreductase activity, acting on paired donors, with incorporation or reduction in molecular oxygen	MF	GO:0016705	19	0.00
chitin catabolic process	BP	GO:0006032	9	0.00

**Table 8 insects-16-01024-t008:** Identification of differentially expressed genes between female and male *Blaptica dubia* by qRT-PCR.

	Gene Name	Log2 Fold Change by RNA-Seq	Log2 Fold Change by qRT-PCR
CSPs	TRINITY_DN5543_c1_g1	−1.22	−4.990831506
	TRINITY_DN23994_c0_g1	−1.80	-0.441161691
OBPs	TRINITY_DN10322_c0_g1	1.88	0.371810946
	TRINITY_DN6009_c0_g1	−1.32	−2.452316478
	TRINITY_DN8952_c1_g1	−1.20	−2.696191085
	TRINITY_DN4687_c0_g1	−1.18	−4.506228313
GRs	TRINITY_DN10421_c0_g1	1.79	−1.728146306
	TRINITY_DN8525_c2_g1	1.41	−1.139610179
ORs	TRINITY_DN2213_c0_g3	1.54	0.249326919
IRs	TRINITY_DN2362_c0_g1	1.49	0.016931829
	TRINITY_DN54802_c0_g1	2.14	0.070340029

## Data Availability

Data are contained within the article. The data presented in this study can be requested from the authors.

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
