# Peer review of "Ultrastructure and Transcriptome Analysis Reveal Sexual Dimorphism in the Antennal Chemosensory System of Blaptica dubia"

_insects, 2025, doi:10.3390/insects16101024_

Round 1
Reviewer 1 Report
Comments and Suggestions for Authors
Dear authors, thank you for your interesting work. This text may still need to be further improved. The structure of the entire text is suitable. It is suggested to make revisions. In addition, the presentation of the work should focus on the distinction between male and female, especially in the transcriptome content, where too much information is presented, often not highlighting important results. There are many tables and figures, and some have obvious repetitions. It would be great if this work could be further improved. For specific suggestions, please refer to the pdf.

There are many mistakes or inaccurate language. It is better that MS was revised by native English speakers.
Author Response
Insects-3838501
Manuscript Title: Ultrastructure and Transcriptome Analysis Reveal Sexual Di-morphism in the Antennal Chemosensory System of Blaptica dubia
Dear editors and reviewers:
We are very appreciative of the hard work undertaken by the academic editor and reviewers on our manuscript (Manuscript ID insects-3838501), Manuscript Title: Ultrastructure and Transcriptome Analysis Reveal Sexual Di-morphism in the Antennal Chemosensory System of Blaptica dubia. We are also truly grateful for the reviewers’ suggestions and comments that have helped improve our manuscript. Based on their feedback, we have carefully considered these suggestions, making thorough revisions to both the content and the language of our original manuscript. The itemized responses to each reviewer’s comments are attached below. The modified parts are marked in red font in the original text. We acknowledge again for your attention to our paper.
Yours sincerely,
Yu Zhang
Below you will find the details of our revision with general reply and point-by-point responses to the reviewers’.
- How to distinguish only the gender differences or sexual dimorphism? It seems that the distinction between the…
Reply: Thanks for your comments. We distinguished between male and female Blaptica dubia based on their wing morphological characteristics (females with short wings, males with long wings). This description has been incorporated into the first sentence of the Abstract in the revised manuscript. Please refer to the modified version for details. (Lines 10-11)
- The abstract seems to redundant. It is suggested that the focus on differeces. The focus is on the …
Reply: Thank you for your valuable feedback on the abstract. We fully agree with your suggestions and have streamlined the abstract to focus specifically on the core findings of sexual dimorphism. Please refer to the revised manuscript for details. (Lines 10-30)
- Simplification.
Reply: Thank you for your valuable feedback. We have streamlined the abstract to reduce redundancy, as detailed in the revised manuscript. (Lines 10-30)
4.The abstract needs to be rewritten.
Reply: Thank you for your feedback. We have revised the abstract to enhance its clarity and focus, with particular emphasis on restructuring the content and highlighting the key findings. The specific modifications are reflected in the updated manuscript. (Lines 10-30)
5.It seems that this sentence has no meaning at all.
Reply: Thank you for your feedback. We have revised the manuscript by removing the original sentence and integrating its key content into the preceding sentence. Please refer to the updated version for details. (Lines 37-39)
- L62, "Blattodea " is meaning order level?No italics.
Reply: Thank you for pointing this out. We have corrected the formatting of "Blattodea" to non-italic. Furthermore, based on the relevant references, we have revised the original description, and the term no longer appears in the corresponding sentence of the current version. (Lines 49-54)
- In Blattodea, the chemosensory system commonly exhibits significant adaptive sexual dimorphism, reflected by divergent sensillar architectures and sex-biased expression of chemosensory genes, add the reference.
Reply: Thank you for your comment. This statement was originally based on the research findings of Periplaneta americana presented in the subsequent sentence, and therefore lacked direct references. Recognizing that extrapolating patterns from Periplaneta americana to the entire order Blattodea may lack rigor, we have revised the text to specifically limit the conclusion to Periplaneta americana. Please refer to the modified manuscript for details. (Lines 49-54)
- "SEM " first appears in the main text and cannot be abbreviated.
Reply: Thank you for your feedback. We have now spelled out the initial occurrence of "SEM" as "Scanning Electron Microscopy" in the manuscript. Please refer to the revised version for specific details of the modifications. (Lines 64-65)
- L107, species name is italic.
Reply: Thank you for pointing this out. We have italicized the scientific names of species. Please refer to the revised version for specific details of the modifications. (Line 95)
- L136-147,I guess this work been done by a biotech company. Why didn't they inform and highly abbreviated?
Reply: We thank the reviewer for their comment. The transcriptome sequencing was performed by LC-Bio Technology Co., Ltd. (Hangzhou). We had originally described the sequencing service in Section 2.5 "Total RNA Extraction and Quality Assessment". To better align with the logical flow of the manuscript, we have now moved this description to Section 2.6 "Transcriptomic Library Construction and Sequencing Protocol". We agree that the original description of the commercial sequencing and analytical procedures was overly detailed. As suggested, we have significantly condensed this section in the revised manuscript. The detailed operational steps have been replaced with a concise summary. Please refer to the revised version for specific details of the modifications. (Lines 124-132)
- L168, Why is there a document here? Isn't the result of own observation?
Reply: Thank you for your comment. The terms "scape," "pedicel," and "flagellum" used to describe the antennal segments of the Blaptica dubia are based on established morphological nomenclature in entomology, rather than being coined by ourselves. The references were cited to indicate the source of this standardized terminology. (Line 152)
- L169-176, the data such (males: 0.49±0.01 mm; females: 0.45±0.01 mm) males: 17.53 ± 0.32 mm; females: 173.
Reply: Thank you for pointing this out. We have removed all instances of "P<0.05" within the parentheses in that section as suggested. Please refer to the revised version for detailed modifications. (Lines 150-160)
- The structure of the text is chaotic and inconsistent.
Reply: Thank you for your feedback. We have revised the section numbering as suggested and changed the heading of section 3.2 to "Transcriptome Sequencing and Data Analysis." The detailed content on transcriptome sequencing has been moved to a subsection under this heading. Please refer to the revised manuscript for specific modifications.
- L262-273, A brief description of this part of the work.
Reply: Thank you for your comment. We have refined the language in this section to improve conciseness and clarity. Please refer to the revised manuscript for specific details of the modifications. (Lines 246-252)
- The work of 3.8 and 3.9, these several parts of work are just a simple accumulation. It is suggested to simplif...
Reply: Thank you for your feedback. We have streamlined and condensed this section for greater conciseness. Please refer to the revised manuscript for specific details. (Lines 257-274)
- Figure 7 and 8, The visibility of the picture is poor.
Reply: Thank you for your valuable suggestions. We have adjusted the images based on the feedback. Since the original Word document was too large to upload directly, we compressed it, which may have reduced the clarity of the images in the document. If you need to view high-definition images, please refer to the original image files we have uploaded together.
- In discussions, it is suggested to deepen the discussion on the association between morphological differences an...
Reply: Thank you for your valuable suggestion. We fully agree with your perspective and have strengthened the discussion on the association between morphological differences and gene expression in the revised manuscript. Please refer to the modified version for specific details. (Lines 424-464)
Reviewer 2 Report
Comments and Suggestions for Authors
Please see the attached review file. Thank you.

Please see the attached review file. Thank you.
Author Response
Insects-3838501
Manuscript Title: Ultrastructure and Transcriptome Analysis Reveal Sexual Di-morphism in the Antennal Chemosensory System of Blaptica dubia
Dear editors and reviewers:
We are very appreciative of the hard work undertaken by the academic editor and reviewers on our manuscript (Manuscript ID insects-3838501), Manuscript Title: Ultrastructure and Transcriptome Analysis Reveal Sexual Di-morphism in the Antennal Chemosensory System of Blaptica dubia. We are also truly grateful for the reviewers’ suggestions and comments that have helped improve our manuscript. Based on their feedback, we have carefully considered these suggestions. We acknowledge again for your attention to our paper.
Yours sincerely,
Yu Zhang
Below you will find the details of our revision with general reply to the reviewers’.
Dear reviewer,
We sincerely thank you for your insightful and constructive comments on manuscript ID insects-3838501. Your keen perspective on academic heritage and regional balance has prompted us to re-examine our research from a more global viewpoint, which has been an invaluable opportunity for our academic growth.
We fully agree with your view that scientific research should transcend geographical boundaries and present academic lineage impartially. Following your guidance, we have thoroughly revisited our reference system, with special emphasis on incorporating pioneering contributions from scholars in North America, Brazil, Europe, and India. In the revised manuscript, out of a total of 41 references, 25 now originate from the regions you emphasized (as detailed in the revised manuscript). The integration of these classical references has indeed enriched and added depth to our discussion.
We once again thank you for guiding us to improve this work with your rigorous academic standards. Your comments have not only enhanced the quality of our study but have also deepened our understanding of academic inheritance. We look forward to receiving your approval of the revised manuscript.
Yours sincerely,
Yu Zhang
Reviewer 3 Report
Comments and Suggestions for Authors
This manuscript integrates scanning electron microscopy (SEM) and antennal transcriptomics to characterize sexual dimorphism in Blaptica dubia. The authors report morphological differences in sensilla and identify sex-biased expression of several chemosensory genes (e.g., OBPs/CSPs enriched in females; GRs/ORs/IRs upregulated in males). The work fills a gap in cockroach sensory biology and provides a foundation for future studies on sex-specific chemical communication in Blattodea.
Overall, the study is relevant, well-written, and mostly methodologically sound. However, authors are required to address the following concerns regarding the manuscript:
Major Concerns:
- The authors have used only five individuals per sex for SEM analysis. This is a small sample for quantitative morphometric comparisons. The authors need to either justify this limitation or expand the sample size.
- The authors need to justify their usage of arginine kinase as a housekeeping gene in the qPCR analysis. Ideally, an additional control, such as a ribosomal protein, would have been better.
- The authors provided P values without explaining the statistical tests performed. The authors need to mention them both in the methods section as a separate subheading as well as in the figure legends. In case the authors have used parametric tests in any case, they need to provide a normality test as a justification.
- qPCR is less sensitive than RNAseq. Yet in some cases, the authors observed stronger observations in qPCR than in RNAseq. How do the authors justify this?
Besides, I have some minor concerns:
- Antenna also contains the Johnston's organ, which has been completely ignored in this manuscript.
- The discussion sometimes over-interprets results. For example, claims that males “evolved reduced dependence on conventional odorant detection” or a “fast-response chemosensory system” are speculative without behavioral evidence. The authors need to temper these statements and clearly distinguish observation from hypothesis. Similarly, the conclusion that females evolved a “high-sensitivity” system is interesting but remains correlative without any behavioral validation. The authors may do a BLAST search for the differentially chemosensory genes with Drosophila in case that yields information regarding their functions.
- The authors need to provide full forms of all abbreviations on their first use.
- The authors can draw comparisons with similar studies conducted on other insects in the discussion section
- The authors need to cite references while explaining the methodologies used
Mostly fine. Some minor typos are there.
Author Response
Insects-3838501
Manuscript Title: Ultrastructure and Transcriptome Analysis Reveal Sexual Di-morphism in the Antennal Chemosensory System of Blaptica dubia
Dear editors and reviewers:
We are very appreciative of the hard work undertaken by the academic editor and reviewers on our manuscript (Manuscript ID insects-3838501), Manuscript Title: Ultrastructure and Transcriptome Analysis Reveal Sexual Di-morphism in the Antennal Chemosensory System of Blaptica dubia. We are also truly grateful for the reviewers’ suggestions and comments that have helped improve our manuscript. Based on their feedback, we have carefully considered these suggestions, making thorough revisions to both the content and the language of our original manuscript. The itemized responses to each reviewer’s comments are attached below. The modified parts are marked in red font in the original text. We acknowledge again for your attention to our paper.
Yours sincerely,
Yu Zhang
Below you will find the details of our revision with general reply and point-by-point responses to the reviewers’.
1.The authors have used only five individuals per sex for SEM analysis. This is a small sample for quantitative morphometric comparisons. The authors need to either justify this limitation or expand the sample size.
Reply: Thank you for your valuable feedback. Regarding the sample size issue, we would like to clarify the following: All cockroaches used in this study were adults from the same batch, reared under identical conditions to ensure consistency in genetic background and physiological status. For the quantitative analysis of antennal morphology, we selected five individuals per sex for measurement. To enhance data reliability, both antennae of each individual were independently prepared and observed, resulting in actual statistical data of 10 antennae per sex. Furthermore, we fully recognize the importance of sample representativeness in morphological research. To address this, we additionally prepared and observed antennae from another set of five male and five female adults, conducting scanning electron microscopy imaging. The data from these supplementary samples have been incorporated into the revised version of the manuscript. The results of this repeated experiment were entirely consistent with the conclusions drawn in the paper, further validating the reliability of our findings. Our sample size determination was also informed by multiple similar studies (e.g.: Scanning Electron Microscopy of Antennae of Coccinella Septempunctata (Coccinellidae: Coleoptera)). In summary, we believe the current data volume is sufficient to support the main conclusions of this study.
2.The authors need to justify their usage of arginine kinase as a housekeeping gene in the qPCR analysis. Ideally, an additional control, such as a ribosomal protein, would have been better.
Reply: Thank you very much for your valuable feedback. We fully agree with the principle that reference genes must be rigorously validated. Although our initially used reference gene (ArgK-Initial) demonstrated stable expression in male and female antennae upon experimental validation, following your suggestion, we sought to adopt a peer-reviewed, species-specific, and primer-validated reference gene. We reviewed literature on reference gene evaluation for our study species (Blaptica dubia) [KarakaÅŸ, E.U.; PektaÅŸ, A.N.; Berk, Åž. Selection and Validation of Potential Reference Genes for Quantitative Real-Time PCR Analysis in Blaptica Dubia (Serville, 1838)(Blattidae, Blaberidae). Cumhuriyet Science Journal 2022, 43, 176-182.]. This study evaluated six candidate genes, explicitly recommending two of them (GADPH and RPS18) as the most stable, while identifying one gene (ACTB) as unstable, and making no definitive conclusions regarding the remaining three genes (α-TUB, EF1α, ArgK).
We synthesized all six primer pairs provided in the literature and tested them in our experimental system (based on Blaptica dubia antennal cDNA). However, we encountered a critical technical challenge: except for the ArgK primer, which was not conclusively recommended in the literature, the other five primer pairs (including the two recommended reference gene primers) failed to produce specific and effective amplification in our cDNA samples (no amplification curve was observed). This strongly indicates, as noted in [KarakaÅŸ et al., 2022], that "the validity of RT-qPCR normalization studies depends on the reference genes included in the study, and the expression stability of these reference genes can be affected by variations in the examined tissues, physiological or experimental conditions. On the other hand, there is no single 'universal' housekeeping gene that is consistently expressed and applicable to all cell and tissue types under various experimental situations."
Given these circumstances and adhering to the above principles, we adopted the ArgK primer from the literature, which was well-designed and demonstrated efficient and specific amplification under our laboratory conditions. The core objective of our study is to compare gene expression between male and female Blaptica dubia antennae. Therefore, we conducted strict validation specifically for this experimental purpose. We confirmed that the ArgK gene exhibits highly stable expression in the two key comparison groups—female antennae and male antennae.
Based on the systematic validation process described above, we have substantial evidence to demonstrate that for the specific purpose of this study—comparing gene expression between male and female Blaptica dubia antennae—the ArgK reference gene we selected is supported by literature, validated through primer testing, and most critically, proven to be stable in the directly compared groups (female vs. male antennae). Thus, it is entirely reliable and justified. (Lines 138-146)
3.The authors provided P values without explaining the statistical tests performed. The authors need to mention them both in the methods section as a separate subheading as well as in the figure legends. In case the authors have used parametric tests in any case, they need to provide a normality test as a justification.
Reply: We sincerely thank the reviewer for raising this important point. In response to the suggestion, we have added a dedicated "Statistical Analysis" subsection in the Methods section, which explicitly states that the total antennal length was measured using six antennae per sex, and sensillum dimensions were obtained by measuring ten distinct structures from different antennal segments. Independent samples t-tests (SPSS 27 software) were employed for intergroup comparisons. All datasets underwent Shapiro-Wilk normality testing and were found to meet the normality assumption (all significance levels > 0.05). Data are presented as mean ± standard error. Furthermore, specific statistical methods have been annotated in the relevant figure captions. These revisions have been incorporated into the revised manuscript, and we greatly appreciate the comment which has enhanced the rigor of our study. (Lines 107-113)
4.qPCR is less sensitive than RNAseq. Yet in some cases, the authors observed stronger observations in qPCR than in RNAseq. How do the authors justify this?
Reply: RNA-Seq indeed exhibits high sensitivity at the limit of detection (LOD), enabling the detection of a greater number of low-abundance transcripts. In contrast, qRT-PCR, due to its focus on single or small-scale gene detection, possesses low background noise, a high signal-to-noise ratio, and operates on the principle of absolute quantification of molecular counts, which makes it perform better in detecting the magnitude of expression changes (|log2FC|). This is not due to insufficient sensitivity of RNA-Seq but rather reflects the different technical characteristics of the two methods. Using qRT-PCR to validate and correct systematic biases in RNA-Seq precisely demonstrates the complementarity between the two technologies, making the research results more accurate and reliable.
5.Antenna also contains the Johnston's organ, which has been completely ignored in this manuscript.
Reply: We thank the reviewer for this insightful comment. The Johnston's organ, an important sensory structure within the antenna, is primarily associated with auditory and vibration perception. As the present study focused specifically on the morphometric analysis of antennal chemosensilla, the experimental design did not include specialized investigation of the Johnston's organ. We have now added a discussion of this point in the revised manuscript, and further details can be found in the modified Discussion section. (Lines 323-329)
6.The discussion sometimes over-interprets results. For example, claims that males “evolved reduced dependence on conventional odorant detection” or a “fast-response chemosensory system” are speculative without behavioral evidence. The authors need to temper these statements and clearly distinguish observation from hypothesis. Similarly, the conclusion that females evolved a “high-sensitivity” system is interesting but remains correlative without any behavioral validation. The authors may do a BLAST search for the differentially chemosensory genes with Drosophila in case that yields information regarding their functions.
Reply: We sincerely thank the reviewer for this important and constructive feedback. You are absolutely correct that our original wording in some instances extended beyond what the data could directly support, conflating correlative observations with speculative hypotheses, and we apologize for this oversight. Following your suggestion, we have removed or substantially revised all overinterpreted statements, adopting more cautious and observation-based language throughout the manuscript. Detailed modifications can be found in the revised version.
We have carefully considered the reviewer's suggestion regarding BLAST homology analysis. After thorough deliberation, we concluded that this analytical approach may not be the most appropriate for our specific study system. Our decision not to perform this analysis was based on the following considerations: the Blattodea species examined in our study are evolutionarily distant from Drosophila, and significant divergence has likely occurred in their chemosensory gene families. Direct comparison presents several limitations: (1) sequence comparisons with low homology may yield substantial false-negative results; (2) even when homologous genes are identified, their functions may have diverged considerably due to the substantial differences in ecological niches and behavioral patterns between these taxa, thereby compromising the reliability of functional inferences.To ensure the rigor of our conclusions, we have chosen to focus our analysis on species-specific gene expression patterns and morphology-function relationships, which already provide substantial evidence for sex-specific adaptations in the chemosensory system.
7.The authors need to provide full forms of all abbreviations on their first use.
Reply: We thank the reviewer for this important reminder. We have now carefully reviewed the manuscript and ensured that all abbreviations (including but not limited to OBP, CSP, IR, GR, OR, etc.) are provided in their full forms upon first occurrence in both the abstract and main text. These modifications have been implemented throughout the revised manuscript.
8.The authors can draw comparisons with similar studies conducted on other insects in the discussion section
Reply: We sincerely thank the reviewer for this valuable suggestion. As recommended, we have now added comparisons with other insect groups in the Discussion section. Specifically, we have incorporated the following content with supporting references:
"This sensory specialization evolved for rapid and accurate conspecific signal recognition is not unique to Blattodea. In moths, the 'balanced olfactory antagonism' mechanism proposed by Baker (2008) similarly reveals how males achieve rapid and precise mate localization in complex chemical environments through specific receptor activation and avoidance of inhibitory signals, reflecting convergent evolution under sexual selection pressures across species." And: "As reviewed by Bruce and Pickett (2011), female insects exhibit fine discrimination capabilities toward blends of plant volatiles during oviposition host selection. This highly sensitive chemosensory system is crucial for accurately assessing the suitability of oviposition sites, which aligns with the 'high sensitivity' strategy implied by the elongated ST2 sensilla and highly expressed OBP/CSP genes in female Blaptica dubia in our study."
These additions have been included on the revised manuscript. The references cited are:
[35] Baker, T.C. Balanced olfactory antagonism as a concept for understanding evolutionary shifts in moth sex pheromone blends. Journal of Chemical Ecology 2008, 34, 971-981.
[37] Bruce, T.J.; Pickett, J.A. Perception of plant volatile blends by herbivorous insects–finding the right mix. Phytochemistry 2011, 72, 1605-1611.
We believe these comparisons significantly strengthen the discussion by placing our findings in a broader evolutionary context. Thank you again for this helpful suggestion. (Lines 426-466)
9.The authors need to cite references while explaining the methodologies used
Reply: Thank you very much for your valuable comments. We fully agree with your suggestion that it is essential to cite relevant literature in the Materials and Methods section to support the methodologies we employed. Following your guidance, we have revised the manuscript by citing three references (Guo et al., 2022; Chang et al., 2015; Li et al., 2018) that closely align with the scanning electron microscopy sample preparation methods used in our study. These references represent widely accepted standard protocols in the field of ultrastructural research on insect antennal sensilla. (Lines 77-100)
Round 2
Reviewer 2 Report
Comments and Suggestions for Authors
Please see the file with my review opinion attached.

Reviewer 3 Report
Comments and Suggestions for Authors
The authors satisfactorily addressed all my comments. They performed additional experiments as required, and when that was not feasible, they provided an appropriate explanation. They also addressed the changes in the text that were suggested. I am satisfied with the current version of the manuscript.